# Recurrent *NOMO1* Gene Deletion Is a Potential Clinical Marker in Early-Onset Colorectal Cancer and Is Involved in the Regulation of Cell Migration

**DOI:** 10.3390/cancers14164029

**Published:** 2022-08-20

**Authors:** Jésica Pérez-García, Abel Martel-Martel, Paula García-Vallés, Luis A. Corchete, Juan L. García, Nerea Gestoso-Uzal, Rosario Vidal-Tocino, Óscar Blanco, Lucía Méndez, Manuel Sánchez-Martín, Manuel Fuentes, Ana B. Herrero, Andreana N. Holowatyj, José Perea, Rogelio González-Sarmiento

**Affiliations:** 1Institute of Biomedical Research of Salamanca (IBSAL), SACYL-University of Salamanca-CSIC, 37007 Salamanca, Spain; 2Molecular Medicine Unit, Department of Medicine, University of Salamanca, 37007 Salamanca, Spain; 3Institute of Molecular and Cellular Biology of Cancer (IBMCC), University of Salamanca-CSIC, 37007 Salamanca, Spain; 4Medical Oncology Department, Complejo Asistencial Universitario de Salamanca-IBSAL, 37007 Salamanca, Spain; 5Hematology Department, Complejo Asistencial Universitario de Salamanca-IBSAL, 37007 Salamanca, Spain; 6Centro de Investigación Biomédica en Red de Cáncer (CIBERONC), 28029 Madrid, Spain; 7Anatomy Pathology Service, University Hospital of Salamanca, 37007 Salamanca, Spain; 8Transgenic Service, Nucleus, University of Salamanca, 37007 Salamanca, Spain; 9Department of Medicine and Cytometry General Service-Nucleus, CIBERONC, Cancer Research Centre (IBMCC/CSIC/USAL/IBSAL), 37007 Salamanca, Spain; 10Proteomics Unit, Cancer Research Centre (IBMCC/CSIC/USAL/IBSAL), 37007 Salamanca, Spain; 11Department of Medicine, Vanderbilt University Medical Center, Nashville, TN 37232, USA

**Keywords:** cell migration, early-onset colorectal cancer, NOMO1, OMICS

## Abstract

**Simple Summary:**

The incidence of EOCRC (age < 50 years at diagnosis) with unknown causes is rising worldwide, necessitating the mechanistical analysis of its molecular basis. The *NOMO1* gene is deleted in a high number of EOCRC tumors compared to LOCRC. In this work, we aimed to test the *NOMO1* gene mutational profile in EOCRC tumors and to characterize the effect of *NOMO1* loss in different CRISPR/cas9-edited cell lines, as well as in murine models. Here, we show that the *NOMO1* gene can be inactivated not only by deletion but also by pathogenic mutations in EOCRC. Our results indicate that *NOMO1* loss could be a passenger mutation in the development of EOCRC, although it contributes significantly to colon cancer cell migration.

**Abstract:**

The incidence of early-onset colorectal cancer (EOCRC; age younger than 50 years) has been progressively increasing over the last decades globally, with causes unexplained. A distinct molecular feature of EOCRC is that compared with cases of late-onset colorectal cancer, in EOCRC cases, there is a higher incidence of *Nodal Modulator 1* (*NOMO1)* somatic deletions. However, the mechanisms of *NOMO1* in early-onset colorectal carcinogenesis are currently unknown. In this study, we show that in 30% of EOCRCs with heterozygous deletion of *NOMO1*, there were pathogenic mutations in this gene, suggesting that *NOMO1* can be inactivated by deletion or mutation in EOCRC. To study the role of *NOMO1* in EOCRC, CRISPR/cas9 technology was employed to generate *NOMO1* knockout HCT-116 (EOCRC) and HS-5 (bone marrow) cell lines. *NOMO1* loss in these cell lines did not perturb Nodal pathway signaling nor cell proliferation. Expression microarrays, RNA sequencing, and protein expression analysis by LC–IMS/MS showed that *NOMO1* inactivation deregulates other signaling pathways independent of the Nodal pathway, such as epithelial–mesenchymal transition and cell migration. Significantly, *NOMO1* loss increased the migration capacity of CRC cells. Additionally, a gut-specific conditional *NOMO1* KO mouse model revealed no subsequent tumor development in mice. Overall, these findings suggest that *NOMO1* could play a secondary role in early-onset colorectal carcinogenesis because its loss increases the migration capacity of CRC cells. Therefore, further study is warranted to explore other signalling pathways deregulated by *NOMO1* loss that may play a significant role in the pathogenesis of the disease.

## 1. Introduction

Colorectal cancer (CRC) is the third-most-common cancer diagnosed in men and women worldwide, estimated to have accounted for 1 in every 10 new cancer cases in 2020 [1]. Despite a reduction in the absolute numbers of patients diagnosed with CRC, there has been an increase in CRC incidence among individuals diagnosed before 50 years of age (early-onset CRC, or EOCRC) that is not well understood [2]. Indeed, the pathogenesis of EOCRC is well characterized among individuals with hereditary CRC. However, the majority (over 80%) of EOCRC cases do not carry a germline mutation associated with cancer predisposition (sporadic EOCRC) [3,4]. Thus, it is important to elucidate the molecular etiologies of sporadic EOCRC to accelerate the translation of research findings into clinical application, and reduce the burden of this disease.

An increasing body of evidence suggests that EOCRC has a clinical, pathologic, and molecular presentation distinct from those of CRCs from cases diagnosed at age 50+ years (late-onset CRC, or LOCRC) [5,6,7,8,9]. For example, from a clinical point of view, sporadic early-onset tumors have a worse prognosis than the late-onset ones: they are more aggressive, develop early metastasis, and therefore are associated with poorer survival. From a molecular point of view, EOCRCs show poor differentiation, signet-ring cells, and mucinous histology, typical features of tumors associated with Lynch syndrome. Moreover, in comparison with LOCRC, these tumors show substantial dissimilarities regarding the CIN pattern [10,11].

Assessment of DNA copy number alterations (CNAs) between EOCRC and LOCRC cases has also yielded intriguing observations for further study. We previously identified a recurrent deletion in the short arm of chromosome 16 (16p13.12-p13.11) that presented alone or in combination with other gains or losses of genetic material, and was more frequent in EOCRC than LOCRC cases (33% vs. 16.3%, respectively) [12]. Importantly, in this chromosomal region was located the *Nodal Modulator 1 (NOMO1)* gene—which was subsequently found to be deleted in 70% of EOCRC cases. However, among late-onset CRCs, only 4.5% of cases carried a *NOMO1* deletion. Together, these findings suggest that *NOMO1* may be a promising molecular feature distinct to EOCRC cases for further molecular study [13].

The *NOMO1* gene encodes a 130 kDa transmembrane protein located in the endoplasmic reticulum that forms a protein complex together with Nicalin (NCLN) and the Transmembrane Protein 147 (TMEM147) that inhibits Nodal signaling during embryonic development [14,15]. Importantly, the Nodal pathway is a signal transduction pathway critical for differentiation during embryonic development, and for the maintenance of pluripotency in human embryonic stem cells [16,17,18,19]. The ligand, Nodal, propagates its signal by binding the type I Activin receptor ALK4/7 (*ACVR1B/ACVR1C*) and type II ACTRIIA/ACTRIIB (*ACVR2A/ACVR2B*) in cooperation with its co-receptor Cripto-1 [16,17,20]. This co-receptor binds ALK4 and promotes the recruitment of ACTRIIA/B, helping form the receptor signaling complex. Once activated, the receptor phosphorylates Smad proteins (P-Smads), which bind to a common intermediate, SMAD4 (Co-Smad), allowing the translocation of the complex to the nucleus, where it regulates the transcription of target genes [20]. Although Nodal is not active in most adult tissues, its re-expression and signaling have been linked to multiple types of human cancers [21]. Nodal signaling maintains the self-renewal capacity of cancer stem cells (CSCs) and promotes the invasiveness of several solid tumors, including melanoma, breast, colon, ovarian, prostate, endometrial, and pancreas tumors [18,21,22,23]. In addition, other complex components, including co-receptor Cripto-1, have been found to be overexpressed in a variety of human tumors and human cancer cell lines, whereas low or undetectable levels of expression were detected in normal adult tissues and in non-transformed normal cell lines [16]. Despite the growing importance of Nodal signaling in carcinogenesis, the role of *NOMO1* in the pathogenesis or progression of CRC, particularly EOCRC, remains unexplored.

In the study presented here, we used CRISPR/Cas9 technology to delete endogenous *NOMO1* in multiple independent cell lines bearing the wild type (WT) gene. We also used a gut-specific conditional knockout (KO) mouse model of *Nomo1* to study subsequent tumor development. Specifically, the characterization of the *NOMO1-KO* clones revealed that *NOMO1* loss did not affect either the Nodal signaling pathway activity or cell proliferation. Importantly, *NOMO1* inactivation deregulated the epithelial–mesenchymal transition (EMT) process and increased CRC cell migration. Deletion of *Nomo1* using in vivo models also did not lead to subsequent tumor development. Together, these findings suggest that *NOMO1* is not a driver of early-onset colorectal carcinogenesis and that other signaling pathways deregulated by its loss may play a relevant role in the pathogenesis of EOCRC.

## 2. Materials and Methods

### 2.1. Human Tissue Samples

In this study, 26 formalin-fixed paraffin-embedded (FFPE) blocks of tumors isolated from EOCRC patients diagnosed at the 12 October University Hospital (Madrid), and at the University Hospital of Salamanca, were included. All tumors were from patients under 50 years of age who had previously shown heterozygous deletion of the *NOMO1* gene [13]. In all cases, clinicopathological data, microsatellite instability (MSI) status, and mismatch repair (MMR) gene mutation status were analyzed. All tumor samples were confirmed to be microsatellite stable (MSS)/sporadic and to carry WT MMR genes.

### 2.2. Cell Lines and Culture Conditions

The colon cancer cell line HCT-116 (age 48) and the human stromal cell line HS-5 (age 30) were acquired from American Type Culture Collection (ATCC). The cell lines were cultured in Dulbecco’s modified Eagle’s medium (DMEM) (Sigma-Aldrich, Darmstadt, Germany) supplemented with 10% fetal bovine serum (FBS) (Sigma-Aldrich) and 1% penicillin/streptomycin (Gibco Life Technologies, Grand Island, NY, USA). All cells were incubated at 37 °C in a 5% CO_2_ atmosphere. The presence of mycoplasma was routinely checked with the MycoAlert kit (Lonza, Basel, Switzerland), and only mycoplasma-free cells were used in all subsequent experiments.

### 2.3. DNA Extraction and DNA Quality Evaluation

DNA was isolated from 10 µm sections from each FFPE block. FFPE sections were treated with a deparaffinization solution (Qiagen, Heidelberg, Germany), and DNA was extracted using the QIAamp DNA FFPE Tissue kit (Qiagen, Heidelberg, Germany) according to the manufacturer’s instructions. The Illumina Custom DNA Panel Reference Guide (Illumina, San Diego, CA, USA) was used to measure the DNA quality by comparing FFPE-gDNA amplification potential with a reference non-FFPE gDNA (QCT). To predict the dilution required for each sample, delta Cq values were calculated using qPCR.

### 2.4. Quantitative PCR (qPCR) and Quantitative Reverse Transcription PCR (qRT-PCR)

For real-time quantification, real-time polymerase chain reaction (RT-PCR) was performed using FastStart Universal SYBR Green Master (ROX) in a StepOnePlus™ Real-Time PCR System (Life Technologies-Invitrogen, Carlsbad, CA, USA). The primers used for gene amplification are detailed in Appendix A. For all cases, a fragment of the *LEMD3* gene (single-copy gene) was amplified from the same DNA sample to be used as internal control. PCR experiments were performed using Applied Biosystems StepOnePlus^®^, and analysis was carried out using RQ Manager^®^ software. Three replicates were used for each PCR reaction. The comparative Ct method (2^−∆∆Ct^) was applied to calculate the relative expression levels of each amplicon using *LEMD3* as the reference gene for normalization. RT-PCR specificity of each PCR reaction was verified by melting curve analysis.

### 2.5. NOMO1 Sequencing 

Optimal-quality DNA samples were used to sequence *NOMO1* within a panel of genes by next-generation sequencing (NGS), according to the manufacturer’s instructions. Briefly, this process included the following steps: *NOMO1* was hybridized with the oligo pool. Unbound oligos were removed. *NOMO1* was extended and ligated with bound oligos. The libraries were amplified. Finally, the libraries were cleaned up by magnetic beads. PCR products were quantified using a Qubit fluorometer (Invitrogen, Carlsbad, CA, USA), and the libraries were normalized at 4 nM in a final pool. Sequencing was performed by a MiSeq System (Illumina, San Diego, CA, USA). Variant Studio Software (Illumina, San Diego, USA) was used for subsequent analysis. Somatic variants with >10% frequency, with a quality score > 500 in the bi-directional sequencing quality filter, and that met the software PASS filter, were reported. To predict the pathogenicity of variants of uncertain significance (VUS), in silico analyses were performed using the prediction programs PolyPhen (Polymorphism Phenotyping), Sift (Sorting Intolerant From Tolerant), and CADD (Combined Annotation Dependent Depletion). 

### 2.6. CRISPR/Cas9-Mediated Generation of NOMO1 Knockout Cells 

Exon 3 and adjacent intronic regions of the *NOMO1* gene were selected as target sequences for the CRISPR-Cas9 design. This is because it is the first exon of the gene where the last nucleotide of its coding sequence encodes a codon with the first two nucleotides of the next exon. Therefore, removing this fragment from exon 3 could produce an early truncated *NOMO1* protein. Three sgRNAs (sgRNA1, sgRNA2, and sgRNA3) were designed with the Spanish National Biotechnology Centre (CNB)-CSIC web tool [24]. Three complementary oligonucleotides corresponding to sgRNAs were designed, including two 4-bp overhang sequences (*NOMO1 UP* and *LOW* oligonucleotides; Appendix A). Each pair of oligonucleotides was phosphorylated, annealed, digested with Bpil enzyme (NEB), and ligated to the pSpCas9(BB)-2A-GFP (PX458), also digested with the same enzyme [25].

Subsequently, 50 μL of *E. coli* DH5α cells was transformed with 2 μL of ligated plasmids. Single colonies were grown, and plasmid DNA was extracted and purified using the Danagene plasmid miniprep kit, as per manufacturer protocol (DANAGEN-BIOTED S.L.). Sanger sequencing (data not shown) confirmed the correct insertion of the sgRNAs into the vector. Cell lines were transfected with a combination of two different *NOMO1* CRISPR-Cas9 KO plasmids (4 μg each) or 8 μg of control CRISPR-Cas9 plasmid (empty PX458). 

Transfection for the HCT-116 cell line was carried out using the Amaxa Cell Line Nucleofector Kit V, the Amaxa Nucleofector device (Lonza) with the T-016 program. The HS-5 cell line was transfected by lipofection, as per the manufacturer’s instructions (PolyJet^TM^ In Vitro DNA Transfection Reagent, SignaGen). Single GFP+ cells were sorted into 96-well plates 48 h after transfection using a BD FACSAria^TM^ III flow cytometer. Isolated clones were expanded in culture over a period of 3 weeks, and then genomic DNA was extracted by the phenol-chloroform method. We confirmed the *NOMO1* status of each clone by PCR using the *NOMO1* exon/intron 3 primers (*Forward*: 5′-CAGTGCTCAGTACCATGTAG-3′; *Reverse*: 5′-GGGAGGAATACAAACCCTC-3′). At least two *NOMO1-KO* clones and two controls (WT cell line or clones generated after transfection with the empty plasmid) were assayed for each cell line. The *NOMO1-KO* clones were also confirmed by qPCR and Western blotting (WB).

### 2.7. Western Blotting

Cells were resuspended in RIPA buffer containing protease inhibitors (Complete, Roche Applied Science, Indianapolis), and the protein concentration was measured using the Bradford assay (BioRad). In all, 30 μg of protein samples were separated on an 8% or 12% SDS-PAGE gel (depending on protein size), transferred to an Immobilon-P membrane (Millipore), and incubated with primary specific antibodies overnight (Appendix A). The following day, secondary antibodies were added and immunoblots were incubated for 1 h at room temperature and developed using enhanced chemiluminescence WB detection reagents (Thermo Fisher Scientific, Waltham, MA, USA). β-actin was used as a loading control.

### 2.8. Cell Viability Assay

An MTT assay was used to quantify cell viability and metabolic activity. Metabolically active cells were marked using 3-(4,5-dimethylthiazol-2-yl)-2,5-diphenyl-2H-tetrazolium bromide (Sigma-Aldrich). A total of 5000 cells/well from the *NOMO1-KO* and control clones of the two cell lines were seeded in a 96-well plate. We measured the viability of the cells by cellular metabolic function at 0 (control), 24, 48, and 72 h, adding to each well 10 μL of MTT and incubating for 1 h at 37 °C. The percentage of growth was determined by statistical analysis with SPSS-IBM software.

### 2.9. Wound Healing Assay

A total of 10^6^ cells per well were seeded in 6-well plates containing three well silicone inserts with two defined cell-free gaps (Ibidi, Fitchburg, WI, USA, Inc.). A minimal concentration of FBS was added to the culture medium to maintain survival but inhibit cell proliferation (2% for HCT-116 and HS-5 cell lines). After cell attachment (24 h), culture-inserts were removed creating a scratch. Each experiment was performed in triplicate. Photos of a determined region of each scratch (three wounds/replica) were taken every 10 min for 48 h using a camera attached to a Mikon ECLIPSE TE-2000-E microscope. The ImageJ^®^ program was used in combination with the MRI Wound Healing Tool to calculate the percentage of migration (shown on the *Y*-axis) for each time reference at 0, 12, 24, and 36 or 48 h (shown on the *X*-axis). The SPSS/IBM software was used to calculate migration percentages for *NOMO1-KO* and control clones. 

### 2.10. Transwell Migration Assay

For the Transwell migration assay, 4 × 10^4^ cells of two HCT-116 and HS-5 NOMO1-KO clones and two control clones were suspended in 300 µL of serum-free DMEM. This cell suspension was added to the upper chamber of a 24-multiwell insert system with an 8 µm pore (SARSTEDT), and 600 µL of DMEM with 10% FBS was added to each lower well. After 24 h of incubation, migratory cells were fixed with 3.7% paraformaldehyde for 5 min and stained with 1% crystal violet solution for 15 min. Microscopy pictures were taken for each well with a 10x objective lens, and the stained area was quantified with ImageJ^®^ software.

### 2.11. RNA Extraction, Microarray Data Analysis, and RNA Sequencing 

Total RNA was extracted from three *NOMO1-KO* clones and three control clones for HCT-116 and HS-5 using RNeasy Mini Kit (Qiagen, Hilden, Germany), according to the manufacturer’s instructions. DNA was removed from the samples using RNase-free DNase Set (Qiagen, Hilden, Germany). RNA integrity was assessed using the Agilent 2100 Bioanalyzer (Agilent, Palo Alto, Santa Clara, CA, USA). 

For the microarray analysis, labeling and hybridizations were performed according to protocols from Affymetrix. Washing and scanning were performed using the Affymetrix GeneChip System (GeneChip Hybridization Oven 645, GeneChip Fluidics Station 450, and GeneChip Scanner 7G). The Robust Multi-Array Average (RMA) procedure was used to quantile normalize, background correct, and log_2_ transform raw microarray data [26] in the oligo R package (v.1.54.1) using a custom cdf reference from BrainArray (v.25.0.0) [27]. Differential gene expression analyses were carried out with the limma package (v.3.46.0). Microarray batch-effects were adjusted through the ComBat function from the sva package.

For RNA sequencing (RNA-seq), 0.5 µg of total RNA was used to construct cDNA libraries with TruSeq stranded total RNA with the Ribo-Zero kit (Illumina, San Diego, CA, USA). Then, cDNA libraries were sequenced with NovaSeq 6000 (paired-end 150bp × 2) with a range of 61.8–78.7 M reads/sample, according to the manufacturer’s instructions. The library was constructed and sequenced by Macrogen Inc. (Macrogen, Seoul, Korea). Adapter sequences and low-quality bases were trimmed using Trimmomatic (v.0.39). Raw and trimmed read quality was assessed with the FASTQC tool (v.0.11.9). The surviving paired reads after trimming were mapped to the hg19 human genome with the STAR aligner (v.2.7.9a). The reference genome sequence (hg19, Genome Reference Consortium GRCh37) and annotation data (v.87) were downloaded from the Ensembl website (https://www.ensembl.org (accessed on 27 January 2022)). Gene level counts were calculated using the union mode from the HTSeq package (v.0.12.4). Gene expression count was normalized using the median of ratios method [28], and differential gene expression was analyzed using the DESeq2 R package (v.1.30.1). Batch-effects were adjusted through the ComBat_seq function from the sva R package (v.3.38.0).

Microarray and RNA-seq data have been deposited into the Gene Expression Omnibus (GEO) (http://www.ncbi.nlm.nih.gov/gds/ (accessed on 10 March 2022)) with the accession number GSE198383. For both assays, the false discovery rate (FDR) was controlled by adjusting *p*-values with the Benjamini–Hochberg method. Genes with an absolute value of the fold change (FC) greater than 1.5 (over-expressed) and lower than −1.5 (under-expressed) were selected for further analysis. Gene expression values of the selected genes were visualized using a heatmap with the pheatmap R package (v.4.0.5). The WebGestalt tool (WEB-based Gene set analysis toolkit) [29] was used for pathway enrichment analyses. Hallmarks standards were used to identify the biological processes that could be affected by the *NOMO1* inactivation. 

### 2.12. Proteome Analysis by Liquid Chromatography-Mass Spectrometry (LC-IMS/MS)

Liquid chromatography–mass spectrometry (LC–IMS/MS) was used to analyze changes in protein expression profiles after *NOMO1* inactivation. Four samples of *NOMO1-WT* and *NOMO1-KO* clones were compared for the HCT-116 and HS-5 cell lines. All purified cells (2000–3000 cells) were processed by lysis solution and phosphates protease inhibitors [30]. In all, 0.5 µg of total protein was reduced with 10 mM dithiothreitol (DTT) + 55 mM iodiacetamide at room temperature for 45 min. Protein was digested with trypsin (1:50 *w*/*w*) at 37 °C for 18 h. Then, the peptide mixture was acidified with 0.1% TFA and desalted with C18 StageTips. Samples were stored at −20 °C until LC–IMS/MS tests were performed. The mass spectrometry proteomics data have been deposited at the ProteomeXchange Consortium (http://proteomecentral.proteomexchange.org (accessed on 4 May 2022)) via the PRIDE partner repository with the dataset identifier PXD033636. For the LC–IMS/MS analysis, a nanoUPLC system (NanoElute, Bruker) was used with a C18 column 15 cm × 75 µm, with 1.6 µm C18 particles (Ion Optics Inc., Waltham, MA, USA) and 120 min gradient (3–50% of ACN at 300 nL/min), coupled to a TimsTOF Pro (Bruker). The TimsTOF Pro was operated in PASEF mode (Parallel Accumulation Serial Fragmentation) using Compass Hystar 5.036.0. Settings for the method using 11 samples per day were as follows: mass range 100 to 1700 m/z, 1/K0 start 0.6 V⋅s/cm^2^ and end 1.6 V⋅s/cm^2^, ramp time 110.1 ms, lock duty cycle 100%, capillary vol. 1600 V, dry gas 3 L/min, dry temp 180 °C, PASEF settings 10 MS/MS scans (total cycle 1.27 sec), charge range 0–5, active exclusion 0.4 min, Schedul Target intensity 10000, intensity threshold 2500, and CID collision energy 42 eV [31]. All raw files were analyzed by MaxQuant v1.6.6.0 software using the integrated Andromeda search engine. Data search was against the Human Uniprot Reference Proteome with isoforms (latest version available) and a separated reverse decoy database using a strict trypsin specificity allowing up to two missed cleavages. The minimum peptide length was set to 7αα’carbamidomethylation of cysteine as a fixed modification, and N-acetylation and oxidation of methionine as variable modifications. The first search for peptide tolerance was started at 70 ppm, and the main search was set at 30 ppm. Single-shot samples were set as “to”, and fractions were set as “from”. For each peptide, the maximum peptide mass (Da) was fixed at 8000. All other global or group parameters were predetermined by MaxQuant. PTMS screening was carried out following PTMScan Direct PTMScan^®^ m (Cell Signaling Technology, Danvers, MA, USA) with the slight modifications described [32,33]. 

### 2.13. Mouse Strains, Adenovirus Injection, and Histological Analysis 

A conditional mouse mutant for *Nomo1* has been previously described [34]. In summary, to avoid *Nomo1* expression in C57BL/6J mouse intestine cells, 8 week old *Nomo1*^flox/flox^ and *Nomo1*^flox/+^ mice were infected with a single injection of 300 MOI of Cre adenovirus (Ad5CMVCre-eGFP) administered into the colon area. The infected mice were named *Nomo1*^flox/flox^
*^CreAdV^* and *Nomo1*^flox/+^
*^CreAdV^*. We used *Nomo1*^flox/flox^, *Nomo1*^flox/+^, and *Nomo1*^+/+^ uninfected mice as a control group. For CCR tumor observation, all groups were euthanized at 20 months old. PCRs were performed using the m*Nomo1*-F- and m*Nomo1*-R-specific primers (Appendix A) of genomic DNA from mice colons to demonstrate whether Cre activity effectively removes *Nomo1* exon 3, achieving homozygous or heterozygous *Nomo1* null alleles. Specifically, the animals were housed according to EU guidelines at the Animal Experimentation Service of the University of Salamanca. All mice received a standard diet and were subjected to a 12 h/12 h light/dark cycle. They were maintained in pathogen-free individual cages, under temperature and ventilation control. After animal necropsy, the digestive tract was removed and fixed in formol 10% solution. All samples were processed into serial paraffin sections and stained with haematoxylin-eosin. Subsequently, a pathological analysis of the digestive tract of all mice was carried out at the Pathology Service of the University Hospital of Salamanca.

### 2.14. Statistical Analysis

Data were analyzed using IBM/SPSS software v26 (SPSS Inc., Chicago, IL, USA). All quantitative data are shown as the mean ± standard deviation (SD). Student’s *t*-test was used to compared differences between two groups when they had a normal distribution (test Kolmogorov–Smirnov; *p*-value > 0.05). A *p*-value ≤ 0.05 was considered statistically significant.

## 3. Results

### 3.1. Quantitative PCR Reveals a Single NOMO Gene

The presence of three highly similar genes, named *NOMO1, NOMO2*, and *NOMO3*, has been previously reported in a region of duplication located on the p-arm of chromosome 16 [35]. These genes encode closely related proteins with a 99.6% homology that may have identical functions. The RT-qPCR method was used to confirm the existence of three *NOMO* genes [36]. A specific pair of oligonucleotides was designed to amplify the *NOMO* genes. Using these primers, a DNA fragment that contains part of exon 4 and part of intron 4 (whose sequence is 100% similar to that of the three reported *NOMO* genes) was amplified. Several oligonucleotides were also designed to amplify and quantify different controls (Appendix A). To further quantify the *NOMO* content, we amplified another fragment of the *NOMO* gene, one which contains part of exon 1, a sequence that is 100% identical across the three reported *NOMO* genes. As an internal control, we amplified part of exon 1 of the single-copy gene *LEMD3* (two alleles), a DNA fragment between exon 2-intron 2 of *RBFOX1*, a gene located in proximity to the *NOMO* genes on the same chromosomal region (two alleles), and a fragment of exon 30 of *PKD1* that has no homology with any pseudogene (two alleles). We also amplified exon 13 of *PKD1**,* which has a pseudogene that contains a 100% homologous sequence (4 alleles), as a duplication control gene and exon 5 of the *STS* gene, located on chromosome X, which has only one allele in males. Our results showed that the amount of *NOMO* DNA amplified by RT-qPCR was similar to single-copy genes *LEMD3*, *RBFOX1*, and exon 30 of *PKD1*, whereas this amount was half that of *PKD1* exon 13 and two times that of *STS* exon 5 (Figure 1).

Next, we compared the three reported *NOMO* gene coding regions and found that they differ in 20 nucleotides. These nucleotides correspond to 13 silent and seven missense mutations (Table 1). Only one of these missense mutations (c.26C>T/p.L9P) has not been included in the dSNP database [37] https://www.ncbi.nlm.nih.gov/snp (accessed on 29 July 2021). The allele frequency of these mutations obtained from the ExAC aggregated population [38] showed that most of the *NOMO WT* alleles have a population frequency near 100%. As the presence of two genes would yield a 50% frequency of the mutated allele, these results strongly suggest that there is a single *NOMO* gene.

### 3.2. NOMO1 Is Frequently Inactivated by Deletion or Mutation in EOCRC

We have previously reported that a high proportion of EOCRCs carry a homozygous deletion of the *NOMO1* gene [13]. To expand upon these findings, we sequenced the *NOMO1* gene in 26 EOCRC tumors. Mutational profiling obtained through next-generation sequencing showed that four of the 26 (15.3%) tumors carried a pathogenic mutation (nonsense mutations) in the *NOMO1* gene. In addition, six VUS were identified and classified by prediction programs as possibly pathogenic and three of the 10 (30%) tumors with heterozygous deletion of the *NOMO1* gene harbored a pathogenic mutation in the remaining allele (Table 2). Together, these results indicate that *NOMO1* is frequently inactivated in EOCRC, either through deletion or through mutation.

### 3.3. CRISPR/Cas9 Technology Efficiently Inactivates NOMO1 

To investigate the consequences of *NOMO1* inactivation, we constructed *NOMO1-KO* cell lines using CRISPR/Cas9 technology. We deleted endogenous *NOMO1* in one cell line derived from EOCRC (HCT-116) and in a bone-marrow-derived non-cancerous mesenchymal cell line (HS-5). Each cell line was transfected with a combination of two plasmids (Appendix A). Cell lines were also transfected with the PX458 empty plasmid. After single-cell sorting of GFP+ cells, clones were expanded and screened by RT-qPCR (Figure 2A), PCR-Sanger sequencing (Figure 2B), and WB (Figure 2C). At least two *NOMO1-KO* clones and two control clones for each cell line were efficiently generated and selected for further studies.

### 3.4. NOMO1 Inactivation Significantly Reduces the Expression of NCLN in Cell Lines

*NOMO1* reportedly forms a protein complex with NCLN and TMEM147, and the complex is located in the endoplasmic reticulum [14,39]. Using an RNA interference approach, previous studies have demonstrated that NCLN and *NOMO1* become unstable and, therefore, the expression of *NOMO1* decreases in the absence of the respective binding partner, suggesting that complex formation has a stabilizing effect [39]. Similarly, it was later shown that TMEM147 expression is strongly reduced in NCLN and *NOMO1* knockdown cells [14]. Here, we found that TMEM147 expression was not affected by *NOMO1* loss (Figure 3A and Appendix A). In contrast, NCLN expression was strongly reduced in *NOMO1-KO* cells across both cell lines tested. As the reduction in NCLN levels was not accompanied by a significative decline in the corresponding mRNA expression levels, as determined by qRT-PCR, these results suggest that protein is reduced by a post-transcriptional mechanism (Figure 3B). 

### 3.5. NOMO1 Inactivation Does Not Affect the Nodal Signaling Pathway Activity or Cell Proliferation 

The *NOMO1*–NCLN–TMEM147 complex has been shown to modulate Nodal signaling in developing zebrafish embryos by an unknown mechanism [14,15,39]. Therefore, we next analyzed the abundance of several members of the Nodal pathway in *NOMO1-KO* and control cells. First, we used WB to quantify the protein expression of receptors ALK4 and ACTRII and the co-receptor Cripto-1 [18,23]. We observed no alterations in the abundance of any of these proteins after *NOMO1* inactivation (Figure 4A). Once activated by the Nodal ligand, the receptors phosphorylate the Smad2 and Smad3 proteins that bind to Smad4 (Co-Smad). We found that Smad4 expression was not affected by *NOMO1* inactivation in any of the cell lines analyzed (Figure 4B). To further examine whether *NOMO1* loss affects the activity of the Nodal signaling pathway, we analyzed Smad2/3 and p-Smad2/3 levels in the presence or absence of recombinant human Nodal (rhNodal) (R&D Systems). Smad2/3 and p-Smad2/3 protein expression was found to be similar in WT and *NOMO1-KO* clones (Figure 4C). This strongly indicates that the Nodal pathway activity is not affected by *NOMO1* inactivation. 

To determine whether *NOMO1* loss affects cell growth rates, WT and *NOMO1-KO* clones were cultured and cell proliferation was measured by MTT assays. No significant differences (*p* >0.05) were found between the growth rates of the *NOMO1-KO* and control clones (Appendix A). 

### 3.6. Gene Expression Profiling in NOMO1-KO Cell Lines

Although our results indicate that *NOMO1* inactivation does not affect Nodal pathway activity, we aimed to test whether *NOMO1* loss affects gene expression profiles. Total RNAs were extracted from *NOMO1-KO* and WT clones and processed for expression microarray and RNA sequencing (RNA-seq). For both analyses, an FC greater than 1.5 was considered for up-regulated genes, and an FC lower than −1.5 for down-regulated genes. 

According to the microarray analysis, 126 genes common to the two cell lines (HCT-116 and HS-5) were found to be deregulated when the four *NOMO1-KO* clones were compared to the four *NOMO1-WT* clones (FC > 1.5; FC < −1.5). A total of 81 genes were up-regulated and 45 genes were down-regulated in *NOMO1-KO* clones, compared to WT (Appendix A; Appendix A). According to the RNA-seq data, 592 genes common to the two cell lines were found to be deregulated (FDR < 0.05) when the six *NOMO1-KO* clones were compared to the six control clones. A total of 108 genes were significantly up-regulated and 213 were down-regulated in *NOMO1-KO* clones, compared to WT (Appendix A; Appendix A). Gene Set Enrichment Analysis (GSEA) revealed the existence of 18 and 20 signaling pathways to be enriched by the deregulated genes associated with *NOMO1* loss in the microarray and the RNA-seq assay, respectively (Appendix A). 

The transcriptional profiling data indicated that *NOMO1*-deficient cell lines presented deregulated genes related to the TNFα (tumor necrosis factor α) pathway, as well as to the inflammatory and blood vessel formation processes, among others. These pathways are often disrupted during inflammatory bowel disease (IBD), which is an important risk factor for the development of CRC. Interestingly, microarray and RNA-seq approaches showed that commonly deregulated genes in *NOMO1-KO* cells perturb the EMT process by deregulating *VCAN, CXCL8, PTX3, EDIL3, LOXL1, BDNF, QSOX1, LUM, PCOLCE, CAPG, POSTN, CCN1, CCN2, CD44, SERPINE2, THBS1*, and *VEGFA*. These genes are also involved in altering cell migration [40,41,42,43,44,45,46,47,48,49]. In addition, an over-representation analysis (ORA) of the genes mentioned above indicated that different biological processes are affected by cell migration (Appendix A). Therefore, altering these two processes could help increase the metastatic capacity of *NOMO1-KO* cells. To identify whether *NOMO1* inactivation affects these processes, we first studied the protein expression of the typical EMT-associated markers E-Cadherin and Vimentin and the expression of β-catenin due to its relationship with cell migration, invasion, and metastasis. However, the protein expression of these markers did not show significant changes when comparing *NOMO1-KO* and control clones (Appendix A). 

Taken together, microarray and RNA-seq approaches showed a differential expression profile and a set of affected signaling pathways by *NOMO1* loss, specially EMT and cell migration. 

### 3.7. Protein Expression Profiling in NOMO1-KO Cell Lines

To elucidate the protein expression profile changes after *NOMO1* inactivation in the two cell lines (HCT-116 and HS-5), an LC–IMS/MS analysis was performed. Four samples of a *NOMO1-WT* and *NOMO1-KO* clones were compared for the two cell lines using 0.5 µg of total protein of each clone. The proteome quantification data of 3227 proteins revealed 357 and 486 deregulated proteins (*p* < 0.05) in HCT-116 and HS-5 *NOMO1-KO* cell lines, respectively. A total of 205 up-regulated and 152 down-regulated proteins were found in the HCT-116 *NOMO1-KO* clone. In HS-5, 182 proteins were up-regulated and 304 proteins were down-regulated in the *NOMO1-KO* clone, compared to WT. When we compared commonly deregulated proteins in all the *NOMO1-KO* clones vs. all control clones, we identified 12 up-regulated and 16 down-regulated proteins across the two cell lines (Appendix A). GSEA revealed the existence of two pathways (apical junction and cholesterol homeostasis) enriched (FDR < 0.05) by the deregulated proteins associated with *NOMO1* loss.

In addition, WB detected and confirmed differentially expressed cell-migration-associated proteins in the LC–IMS/MS analysis (CTND1, LMNB1, and HMGA1) (Figure 5). These results justify the alteration of the processes observed in the transcriptome (EMT and cell migration) and suggest that *NOMO1* loss could influence the migratory capability in CRC cells. 

### 3.8. NOMO1 Inactivation Promotes Cell Migration

To determine whether *NOMO1* has an important role in cell migration, we tested the migration capacity of colon cancer cells in the absence of this gene. We performed wound healing and Transwell migration assays to compare the cell migration ability of WT and *NOMO1-KO* clones in both cell lines. For the HCT-116 cell line, the recolonized area was calculated at 24 and 48 h. For the HS-5 cell line, it was calculated at 24 and 36 h (in the wound healing assay), since at 36 h, all the wounds were closed. Consistent across all cell lines, our results showed that the percentage of migration was significantly higher in the *NOMO1-KO* clones compared to controls (t-test, *p* < 0.01) (Figure 6). Therefore, *NOMO1* loss promotes cell migration.

### 3.9. Nomo1 Deficiency in Mouse Colon Cells Does Not Modify the Susceptibility of Developing EOCRC

To see the effect of the somatic deletion of *NOMO1* in EOCRC, and considering that this gene could be a tumor suppressor, we decided to silence *Nomo1* expression in the mouse colon cells using a conditional mouse model [34]. The selection of a conditional mouse model allowed us to induce a *Nomo1* deletion at a specific time and in a specific cellular context. A total of nine mice (8 weeks old) were colon injected with a Cre adenovirus: 6 *Nomo1*^flox/flox^, 2 *Nomo1*^flox/+^, and 1 *Nomo1*^+/+^. As a control group, we used 2 *Nomo1*^flox/flox^ and 1 *Nomo1*^flox/+^. All mice were monitored for tumor development until 20 months after Cre activation, when they were euthanized for histological analysis of the gastrointestinal tract. PCR was used to check for *Nomo1* exon 3 deletion in genomic DNA from mice of experimental and control groups. *Nomo1*^flox/flox;^ *^CreAdV^* and *Nomo1*^flox/+;^ *^CreAdV^* showed a fragment of 482 bp corresponding with the *Nomo1* exon 3 ablation (Appendix A). The pathological study of both transduced and control mice showed the absence of lesions in the digestive tract compatible with the development of CRC or other tumors. This result indicates that *Nomo1* deficiency could not be a driver for the development of CRC tumors in mice.

## 4. Discussion

The rising burden of EOCRC with unknown etiologies is a global epidemic—particularly given that EOCRC is characterized by more aggressive phenotypes and poorer prognostic outcomes, compared with late-onset disease [10]. Despite initial advances in understanding the distinct disease burden (with clinical, pathological, and molecular phenotypes), few studies have explored the mechanistic underpinnings of this malignancy [50,51,52]. Such studies of EOCRC etiology may help translate research findings into clinical advances to improve outcomes specifically for this growing population. Our study is one of the first mechanistic studies of EOCRC biology with a focus on the role of *NOMO1* in colorectal carcinogenesis among young individuals. We applied the CRISPR/Cas9 technology to delete endogenous *NOMO1* in multiple independent cell lines, and used a *NOMO1* gut-specific conditional mouse model to study subsequent tumor development. Characterization of the *NOMO1-KO* clones revealed that though *NOMO1* loss did not affect Nodal signaling pathway activity or cell proliferation, it increased CRC cell migration. There was also no subsequent tumor development on deleting *Nomo1* using in vivo models. Together, these findings suggest that other signaling pathways deregulated by the loss of *NOMO1* may play a relevant role in the pathogenesis of EOCRC.

Until now, three *NOMO* genes had been described, *NOMO1*, *NOMO2*, and *NOMO3,* with a sequence homology of more than 96%. In this work, we observed that the amount of *NOMO* gene amplified by qPCR was the same as that amplified for different single-copy genes. These results, together with (i) an observed frequency of *NOMO1 WT* alleles close to 100% and (ii) the finding that across different species, including the mouse, there is only one *NOMO* gene, strongly suggest the existence of a single *NOMO* gene. Thus, it is possible that alternative *NOMO* alleles are either a rare mutation or a sequence annotation error.

In an earlier study, we reported that more than 70% of EOCRCs examined had lost the *NOMO1* gene [13]. In this study, we expanded upon these findings to investigate the *NOMO1* mutational profile in EOCRC and found that 15.3% of the samples presented a mutation that generated a premature stop codon, resulting in a truncated protein. In addition, six unknown clinical significance variants were identified and classified as probably pathogenic. Thus, we demonstrated that in early-onset colorectal tumors, this gene can be inactivated not only by deletion but also by mutation, which highlights the recurrence of *NOMO1* inactivation in the pathogenesis or progression of EOCRC. To date, the TCGA-COAD database has reported only 3.4% of the *NOMO1* gene mutations in EOCRC, 4.5-fold lower than that observed in our study. This higher incidence of *NOMO1* gene mutations observed in our cohort could be associated with its clinicopathological characteristics, where 90% of the tumors had MSS [13]. This mutational profile could also be related to the environmental influence, given that EOCRC has been shown to exhibit disparities in etiology according to race, sex, and geographic area [53]. Therefore, our data show the existence of *NOMO1* gene pathogenic mutations associated with EOCRC and suggest that this gene may function as a tumor suppressor.

*NOMO1* forms a protein complex with NCLN and TMEM147, and the complex is located in the endoplasmic reticulum and inhibits Nodal signaling—a signal transduction pathway that maintains pluripotency in human embryonic stem cells [14,39]. Previous studies have shown that the formation of the *NOMO1*–NCLN–TMEM147 protein complex is limited by a post-transcriptional regulation mechanism that originates from the assembly of the complexes [14]. The incorporation of these monomeric components in the protein complex stabilizes them, significantly increasing their half-lives [39]. Although the assembly mode of the complex is not well understood, Dettmer et al. (2010) showed the critical role that NCLN plays in the formation of the protein complex by controlling the cellular steady-state levels of *NOMO1* and TMEM147, which are synthesized in excess and their monomeric forms subjected to rapid proteolytic degradation [14]. The inactivation of any of the three proteins resulted in a strong reduction in the other two. However, *NOMO1* and TMEM147 overexpression, alone or in combination, did not yield increased levels of NCLN expression, suggesting that NCLN may be the limiting factor in this protein complex formation [14,39].

In this work, we also observed that *NOMO1* inactivation strongly reduced NCLN protein expression, while the levels of TMEM147 were not modified in any of the cell lines studied herein. These results confirm protein complex destabilization by the modification of *NOMO1* expression levels. However, our results are inconsistent with the reported reduction in TMEM147 levels by *NOMO1* inactivation [14]. We hypothesize that although the protein complex is not correctly assembled, the unbound TMEM147 is not degraded by the proteasome, potentially due to its stability in the monomeric form across the analyzed cell lines. Microarrays, RNA-seq, and qRT-PCR analyses also showed that the destabilization of NCLN in the absence of *NOMO1* was not accompanied by a reduction in its mRNA. These results indicate that protein reduction is caused by a post-transcriptional mechanism, in agreement with the data published by Dettmer et al. [14].

After elucidating the effect of *NOMO1* loss in the *NOMO1*–NCLN–TMEM147 complex, we evaluated its role in activating the Nodal signaling pathway, essential for cellular differentiation during embryonic development [20] and reactivation in multiple tumor types [21], where it has been related to increased proliferation and invasion [18,54]. We found that *NOMO1* inactivation did not perturb the expression of any proteins involved in the Nodal signaling pathway, including both p-Smad2 and p-Smad3, whose levels should be elevated upon signaling pathway activation. These results suggest that the possible carcinogenic effect of *NOMO1* loss is not related to its described role as an inhibitor of the Nodal pathway. Further investigation is essential to identify other biological pathways that might be regulated by *NOMO1* inactivation in EOCRC.

To determine the role of *NOMO1* in EOCRC, we analyzed the transcriptional and protein expression profiles of WT and *NOMO1-KO* cell lines. Our results revealed a differential expression profile of genes commonly deregulated across the HCT-116 and HS-5 cell lines due to *NOMO1* inactivation, with the involvement of several genes associated with cell migration, invasion, and EMT processes. For this reason, we first studied the protein expression of E-Cadherin, Vimentin, and β-Catenin. However, these markers did not show significant changes after *NOMO1* inactivation, suggesting that other proteins may be involved in the deregulation of EMT and cell migration. In this sense, LC–IMS/MS analysis revealed a set of proteins communally deregulated, and associated with migration, invasion, and EMT processes, and that could respond to *NOMO1* inactivation by contributing to the increased migratory capacity.

For example, CTND1 (under-expressed in both cell lines) has been associated with the stabilization of E-Cadherin and the maintenance of cell adherent junctions [55], consistent with the deregulated pathways associated with our proteome analysis. Moreover, CTND1 is involved in the Wnt/beta-catenin/CTNNB1 signaling pathway by regulating cell proliferation, migration, and differentiation of endothelial cells in tumor growth [56]. According to other studies, CTND1 loss is associated with poor prognosis and metastasis in ductal breast cancer and is considered a tumor suppressor [57]. In head and neck squamous carcinoma (HNSCC) and oral squamous cell carcinoma (OSCC) cell lines, loss of CTND1 increases migration and invasion and favors EMT [58,59]. Lamin B1 (LMNB1), under-expressed in both cell lines in our study, has been described as a tumor suppressor in lung cancer. LMNB1 knockdown in lung epithelial cells promoted EMT, cell migration, tumor growth, and metastasis by activating RET/p38 signaling [60]. HMGA1, up-regulated in NOMO1-KO cells, is overexpressed in cervical cancer tissues and is positively correlated with lymph node metastasis and advanced clinical stage. In addition, HMGA1 overexpression enhances tumor growth and accelerates cell migration and invasion in cervical cancer cell lines [61].

Previous studies have shown that Nodal expression can increase invasive and metastatic capacity of tumor cells [21]. Although we did not find increased activation of the Nodal signaling pathway upon *NOMO1* loss, we investigated the effect of *NOMO1* inactivation on cellular migration capacity due to the alteration of the cell-migration-associated proteins (CTND1, LMNB1, and HMGA1), discussed above. Strikingly, we found that *NOMO1* inactivation increased the migration ability of all cell lines analyzed. Taken together, these data provide a possible explanation for the increased migration capacity exhibited by the *NOMO1-KO* clones, and further study is warranted on the role of these genes in EOCRC development. Although increased migration induced by *NOMO1* loss could contribute to colorectal carcinogenesis, we cannot exclude the possibility of other signaling pathways deregulated by *NOMO1* deficiency playing a relevant role in the pathogenesis of the disease.

To study the role of *NOMO1* in colorectal carcinogenesis, a conditional *Nomo1* mouse model was used to analyze this gene function in a rigorous and specific way that avoids embryonic lethality [34]. This animal model, which uses the Cre-*loxp* system to delete a specific gene in a particular tissue, has been efficiently tested by other groups [62,63]. For example, Huang et al. used this method to study tumor formation in the colon [64]. Our in vivo models indicate that the loss of *Nomo1* does not directly contribute to the development of colorectal tumors in mice after 20 months of follow-up. In this case, we think that the cellular and environmental influence on the mouse colon cells differ from the cellular context in human colon during colorectal carcinogenesis.

## 5. Conclusions

In conclusion, our data suggest that, despite being deleted in 70% of EOCRC cases [13] and promoting colon cancer cell migration, *NOMO1* might play a secondary role in the development of this disease. The data also suggest that other coding or non-coding genes located in the same chromosomal region (16p13.11-13.12) could act as driver genes in early-onset colorectal carcinogenesis. Consequently, further mechanistic studies are needed to understand the role of this putative tumor suppressor in EOCRC and the distinct mechanistic underpinnings of early-onset colorectal carcinogenesis.

## Figures and Tables

**Figure 1 cancers-14-04029-f001:**
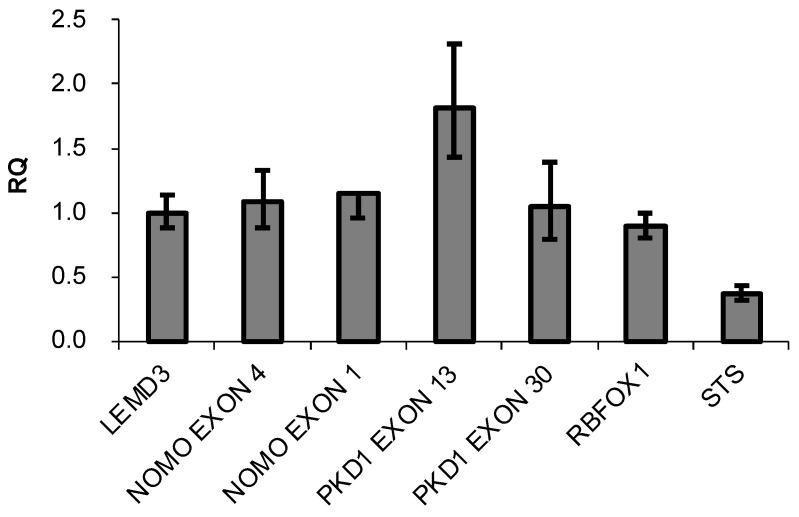
DNA content of the indicated genes determined by q-PCR. Data are presented as the mean of three replicates ± SD. RQ represents the relative expression levels of each amplicon normalized with a control (*LEMD3*).

**Figure 2 cancers-14-04029-f002:**
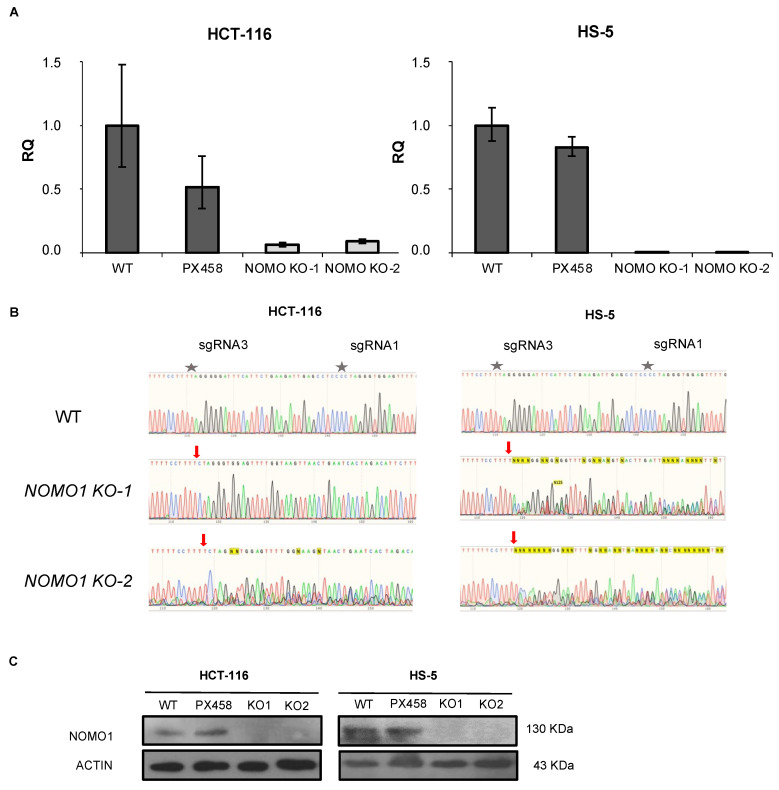
Generation of *NOMO1-KO* cell lines using CRISPR/Cas9 technology. (**A**) qPCR amplification of the fragment corresponding to the sgRNAs used in *NOMO1-KO* and control clones. The RQ was calculated according to the 2^−ΔΔCT^ method, using the WT clones for normalization. Data are shown as the mean ± SD of three replicates. (**B**) Confirmation by Sanger sequencing of the reading frame change in the nucleotide sequence generated by Cas9 in *NOMO1-KO* clones. A red arrow marks the Cas9 breakpoint guided by sRNA1/sgRNA3. (**C**) *NOMO1* expression by WB in WT and KO clones in HCT-116 and HS-5 cell lines.

**Figure 3 cancers-14-04029-f003:**
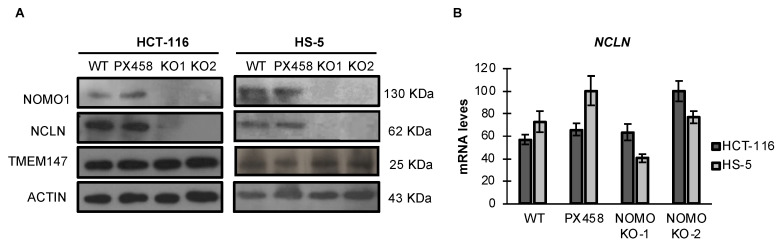
Expression levels of Nicalin and TMEM147 in the presence or absence of *NOMO1* across cell lines. (**A**) *NOMO1*, NCLN, and TMEM147 protein levels, determined by WB. (**B**) mRNA expression of Nicalin determined by qRT-PCR in *NOMO1-KO* and control clones. The expression of each clone was determined using the 2^−ΔΔCT^ method, and GAPDH was used for normalization. Data are shown as the mean ± SD of three replicates.

**Figure 4 cancers-14-04029-f004:**
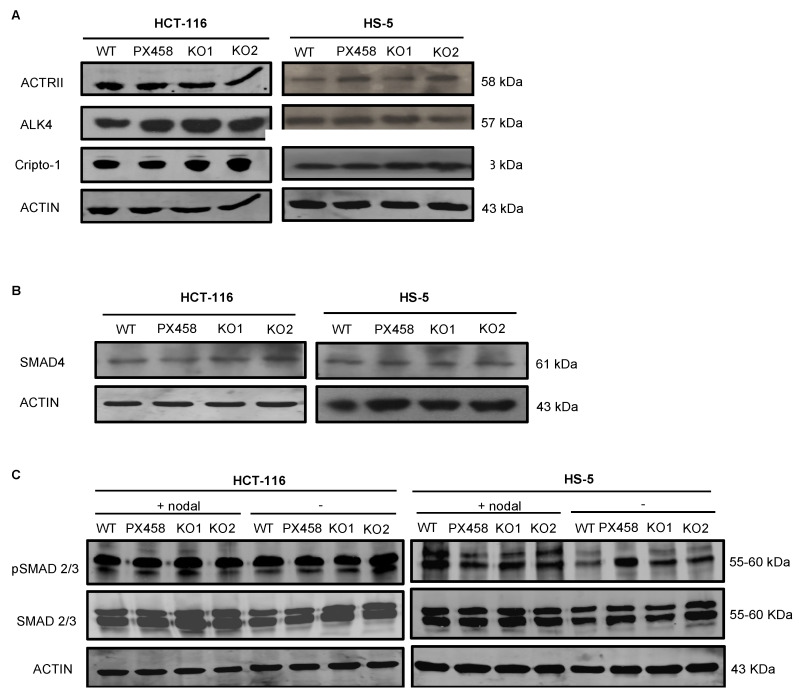
Expression levels of proteins involved in the Nodal signaling pathway in *NOMO1*-*KO* and control clones. (**A**) Expression of ALK4 and ACTRII Nodal pathway receptor proteins and the co-receptor Cripto-1 detected by WB. (**B**) Protein expression of SMAD4. (**C**) Levels of Smad2/3 and p-Smad2/3 protein expression in untreated clones or clones treated with rhNodal (300 ng/mL) for 24 h.

**Figure 5 cancers-14-04029-f005:**
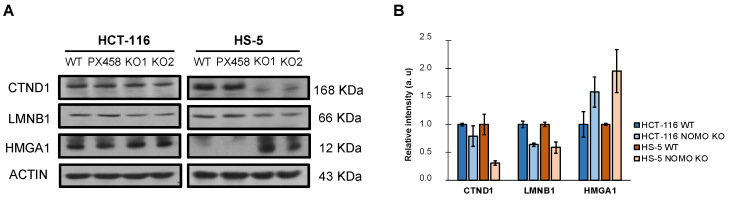
Expression of differentially expressed cell-migration-associated-proteins (CTND1, LMNB1, and HMGA1) in the LC–IMS/MS analysis of the HCT-116 and HS-5 cell lines. (**A**) Western blot analysis showed the differential expression of CTND1, LMNB1, and HMGA1 in *NOMO1* knockout cell lines. (**B**) The graph shows the normalized quantification of CTND1, LMNB1, and HMGA1, detected by Western blot, named as relative intensity in arbitrary units (au).

**Figure 6 cancers-14-04029-f006:**
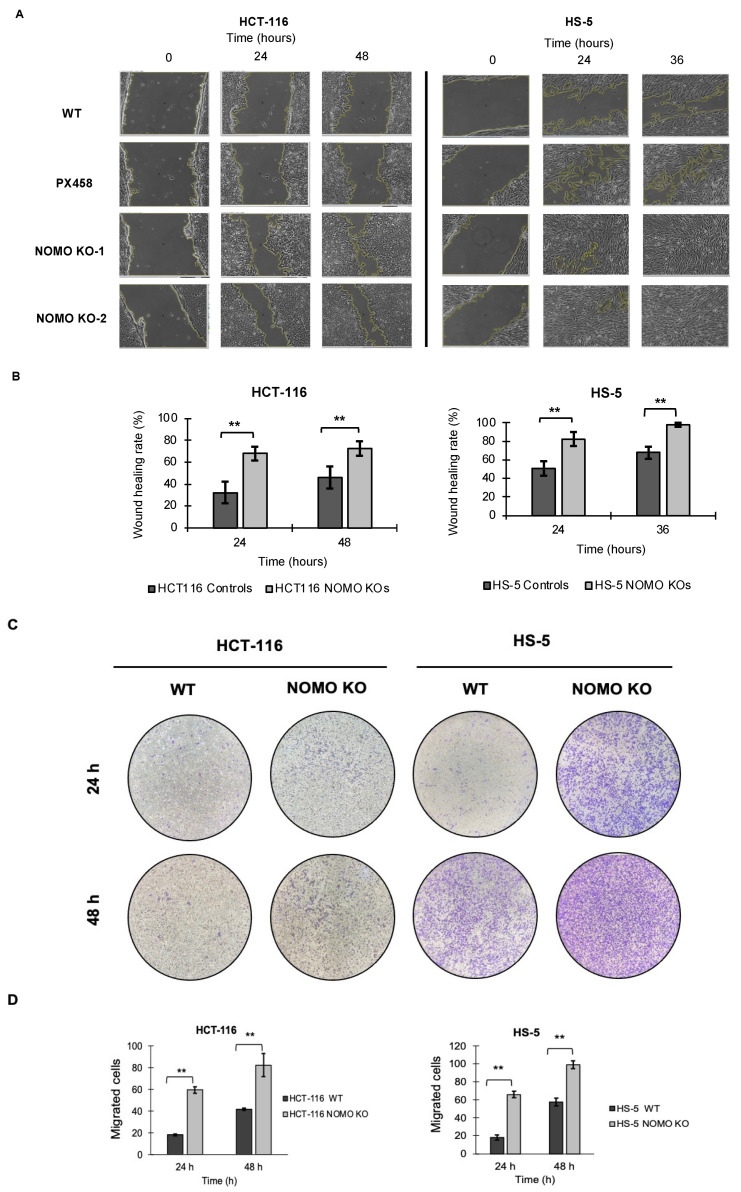
Loss of *NOMO1* promotes cell migration of HCT-116 and HS-5 cell lines. (**A**) Wound healing assay of *NOMO1-KO* and control clones analyzed at 0, 24, and 36 or 48 h. (**B**) Wound healing rate (%) representation for *NOMO1-KO* and control clones. Data are shown as the mean ± SD of the three replicates. Statistically significant differences are indicated by asterisks (** *p* < 0.01). (**C**) Transwell assay assessed at 24 and 48 h of *NOMO1-KO* and control clones. Migrated cells were stained with 1% crystal violet for quantification. (**D**) Quantification of cells with migration capability represented in the Transwell assay as cells migrated to the two cell lines. Data are shown as the mean ± SD. Statistically significant differences are indicated by asterisks (** *p* < 0.01).

**Table 1 cancers-14-04029-t001:** Allele frequency of nucleotides that differ among the reported *NOMO1, NOMO2,* and *NOMO3* genes. Missense mutations are marked in bold.

GVSc	*NOMO1*	*NOMO2*	*NOMO3*	Poblation ID	Subjects	FREQ. ALE>	FREQ. ALE<
**c.26AC > T p.(L9P)**	**C**	**T**	**C**				
c.156T > G p.(S52S)	T	G	T				
c.696C > G p.(N232N)	C	G	C	ExAc	7964	C = 0.997	G = 0.003
c.1185G > A p.(P395P)	G	G	A	ExAc	121,402	G = 0.999	A = 0.001
**c.1210 A > G p.(I404V)**	**A**	**G**	**G**	**ExAc**	**121,400**	**A = 0.999**	**G = 0.001**
**c.1238 A > G p.(Q413R)**	**A**	**G**	**A**	**ExAc**	**11,862**	**A = 0.999**	**G = 0.000**
c.1260C > G p.(P420P)	C	G	C	ExAc	25,306	C = 0.999	G = 0.001
**c.1374 T > G p.(K458N)**	**T**	**T**	**G**	**ExAc**	**27,102**	**T = 0.777**	**G = 0.223**
**c.1468A > G p.(N490D)**	**A**	**G**	**G**	**ExAc**	**23,336**	**A = 0.943**	**G = 0.056**
**c.1477 A > G p.(M493V)**	**A**	**G**	**G**	**ExAc**	**121,206**	**A = 0.999**	**G = 0.001**
**c.1738 A > G p.(M580V)**	**A**	**G**	**G**	**ExAc**	**119,700**	**A = 0.989**	**G = 0.010**
c.2187C > T p.(G729G)	C	T	T	ExAc	121,396	C = 0.999	T = 0.001
c.2211C > T p.(P737P)	C	C	T	ExAc	814	T = 0.814	C = 0.185
c.2388T > C p.(H796H)	T	C	C				
c.2586G > A p.(A862A)	G	A	A				
c.2694C > T p.(S898S)	C	T	T	ExAc	121,402	C = 0.999	T = 0.001
c.3216A > G p.(T1072T)	A	G	G	ExAc	121,310	A = 0.999	G = 0.001
c.3318C > T p.(D1106D)	C	T	T	ExAc	121,406	C = 0.999	T = 0.001
c.3583C > G p.(R1195R)	C	G	G				
c.3666T > Gp.(T1222T)	T	G	G				

**Table 2 cancers-14-04029-t002:** Pathogenic mutations and variants of uncertain significance (VUS) identified in EOCRC tumors. Classification of VUS by different prediction programs is shown: Sift (scores ≤ 0.05 are called “deleterious”, and scores > 0.05 are called “tolerated”), Polyphen (scores > 0.446 are called “probably damaging”, and scores ≤ 0.446 are called “benign”), and CADD (scores > 30 are called “likely deleterious”, and scores ≤ 30 are called “likely benign”). The text in bold indicates pathogenic mutations.

SAMPLE	*NOMO1* Status	Variant	Alellic Frecuency (%)	Sift	Polyphen	CADD Score
**Sample 1**	**Heterozygous**	**c.2684T > A/p.(Leu895Ter)**	**22.5**	**-**	**-**	
**Sample 2**	**Heterozygous**	**c.3637C > T/p.(Gln1213Ter)**	**27.3**	**-**	**-**	
**Sample 3**	**Homozygous**	**c.2833C > T/p.(Gln945Ter)**	**10.5**	**-**	**-**	
**Sample 4**	**Heterozygous**	**c.2428G > T/p.(Glu810Ter)**	**17.2**	**-**	**-**	
Sample 5	Germinal	c.2656G > A/p.(Asp886Asn)	33.3	deleterious (0.02)	probably damaging (0.835)	25.8
Sample 6	Germinal	c.2787G > A/p.(Met929Ile)	16.2	deleterious (0.01)	probably damaging (0.783)	26.4
Sample 7	Homozygous	c.2794G > A/p.(Glu932Lys)	32.1	deleterious (0)	probably damaging (0.977)	32.0
Sample 1	Heterozygous	c.2810C > T/p.(Pro937Leu)	52.9	deleterious (0)	probably damaging (1)	29.5
Sample 3	Homozygous	c.2278G > A/ p.(Gly760Arg)	11.5	deleterious (0.01)	probably damaging (0.992)	29.3
Sample 3	Homozygous	c.3019G > A/ p.(Gly1007Arg)	25.0	deleterious (0)	probably damaging (1)	26.0

## Data Availability

The datasets generated and/or analyzed during the current study are available in the GEO (ID GSE198383; http://www.ncbi.nlm.nih.gov/gds/ (accessed on 10 March 2022)) and PRIDE (PXD033636; http://proteomecentral.proteomexchange.org (accessed on 4 May 2022)) repositories.

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
