# Peer review of "Recurrent NOMO1 Gene Deletion Is a Potential Clinical Marker in Early-Onset Colorectal Cancer and Is Involved in the Regulation of Cell Migration"

_cancers, 2022, doi:10.3390/cancers14164029_

Round 1

Reviewer 1 Report

This is an interesting study in the new area of Molecular tumor biology leading to more research for clinical evaluation.

Author Response

Point 1: This is an interesting study in the new area of Molecular tumor biology leading to more research for clinical evaluation.

Response 1: We thank the Reviewer for this comment.

Reviewer 2 Report

Authors has revealed that NOMO1, a gene that is specifically deleted or inactivated in EOCRCs, contributes to cancer cell migration. The manuscript was informative and persuasive. I have few minor comments:

1. Does the title accurately describe the manuscript? The more mechanistic aspect of NOMO1 is described than just a 'clinical marker'. 

2. Can Fig.5 Western blot analysis results be quantified? The upregulation of HMGA1 in HS-5 is clear but that in HCT-116 is not so clear.

Author Response

Authors has revealed that NOMO1, a gene that is specifically deleted or inactivated in EOCRCs, contributes to cancer cell migration. The manuscript was informative and persuasive. I have few minor comments:

Point 1: Does the title accurately describe the manuscript? The more mechanistic aspect of NOMO1 is described than just a 'clinical marker'.

Response 1: We appreciate this suggestion from the Reviewer. The manuscript tittle has been changed to “Recurrent NOMO1 Gene Deletion is a Potential Clinical Marker in Early-Onset Colorectal Cancer and is involved in the regulation of cell migration”.

Point 2: Can Fig.5 Western blot analysis results be quantified? The upregulation of HMGA1 in HS-5 is clear but that in HCT-116 is not so clear.

Response 2: We thank the Reviewer for raising this point. Fig.5 western blot analysis has been quantified and added to the main text as Fig 5.B.
